# Use of Refractance Window Drying as an Alternative Method for Processing the Microalga *Spirulina platensis*

**DOI:** 10.3390/molecules28020720

**Published:** 2023-01-11

**Authors:** Neiton C. Silva, Luis V. D. Freitas, Thaise C. Silva, Claudio R. Duarte, Marcos A. S. Barrozo

**Affiliations:** Faculty of Chemical Engineering, Federal University of Uberlândia, Block K, Campus Santa Mônica, Uberlandia 38400-902, MG, Brazil

**Keywords:** *Spirulina platensis*, refractance window drying, bioactive compounds, phycocyanin, pharmaceutical potential, anti-cancer products

## Abstract

Microalgae such as *Spirulina platensis* have recently attracted the interest of the pharmaceutical, nutritional and food industries due to their high levels of proteins and bioactive compounds. In this study, we investigated the use of refractance window (RW) drying as an alternative technology for processing the microalga Spirulina biomass aiming at its dehydration. In addition, we also analyzed the effects of operating variables (i.e., time and temperature) on the quality of the final product, expressed by the content of bioactive compounds (i.e., total phenolics, total flavonoids, and phycocyanin). The results showed that RW drying can generate a dehydrated product with a moisture content lower than 10.0%, minimal visual changes, and reduced process time. The content of bioactive compounds after RW drying was found to be satisfactory, with some of them close to those observed in the fresh microalga. The best results for total phenolic (TPC) and total flavonoids (TFC) content were obtained at temperatures of around 70 °C and processing times around 4.5 h. The phycocyanin content was negatively influenced by higher temperatures (higher than 80 °C) and high exposing drying times (higher than 4.5 h) due to its thermosensibility properties. The use of refractance window drying proved to be an interesting methodology for the processing and conservation of *Spirulina platensis*, as well as an important alternative to the industrial processing of this biomass.

## 1. Introduction

*Spirulina platensis* is a kind of blue-green cyanobacterium microalga whose recognized nutritional qualities and nutraceutical properties have been recently studied by many researchers worldwide. This microalga contains a very high amount of protein, which can reach 55 to 70% of its weight in dry mass, including all the essential amino acids and small quantities of methionine, cystine and lysine [1,2,3,4,5]. Spirulina still has about 10 types of vitamins (such as A, E, K, B1, B2, B6 and specially B12) and minerals, such as potassium, iron, calcium, phosphorus, manganese, copper, zinc and magnesium. Its pigments include beta-carotene, chlorophyl and phycocyanin, the last of which is an antioxidant and anti-inflammatory compound used in the cosmetic and pigment industries. In addition, some studies have indicated that a series of phenolics compounds can be found in this microalga as caffeic, chlorogenic, salicylic, synaptic and trans-cinnamic acids, which act individually or synergistically with other antifungal and antioxidant compounds [6,7,8,9,10].

Despite this vast potential, the fresh Spirulina needs to be submitted to a preservation technique because of its elevated moisture content and chemical composition that make it highly perishable [6,8,11]. The literature reports various dehydration techniques that may be used in microalgae, such as convective drying, solar drying, spray drying, cross-flow drying, rotary drum drying and vacuum shelf drying. However, these methods do not generally maintain the quality of the final product and/or add high costs to the process, reinforcing the need for studies on energy-efficient drying systems to better preserve the aspects of the dried product, especially the content of bioactive compounds [12,13,14,15,16].

In this context, refractance window (RW) drying appears as an alternative technique to process the *Spirulina platensis.* In this methodology, the material is spread on the upper face of a flexible support (commonly Mylar^®^ films), while its lower surface is in contact with a hot fluid (e.g., water). This system is commonly used for converting liquid, viscous suspensions, pasty foods and other related biomaterials into films, powders, flakes, or sheets with added value, and it has been used effectively for dehydrating heat-sensitive materials similar to microalgae [17,18,19].

The three modes of heat transfer (conduction, convection, and radiation) are active during the RW drying. The heated fluid, commonly hot water up to 98 °C, is the source of thermal energy in the RW that is transmitted through the Mylar^®^ by conduction and radiation. Thus, when the product with high moisture is spread over the film, the refractive index between the water and material becomes closer, reducing the reflection at the interface and increasing the transmissivity of radiant energy to the product, forming a “window” through which the radiation crosses. Once Mylar^®^ is a low heat conductor, the thermal damage is minimized in the drying steps where the material contains less moisture and the overheating is avoided. In general, the temperatures to which the material is subjected during the initial and intermediate stages of drying are about 20 to 25 °C lower than those of the hot water, due to the effects of evaporative cooling and the transfer of heat by convection between the material and the surrounding air [18,19,20]. Compared to freeze-drying, which is a technique especially used in the case of heat-sensitive raw materials, RW has smaller processing times and higher energy efficiency [18,20,21]. In addition, the quality parameters of RW-dried products (e.g., color and retention of antioxidant compounds and vitamins) are close to those of freeze-dried products, which is a recognized technique for preserving the original properties of the material. The refractance window has been successfully applied to process a wide variety of biomaterials, including strawberries [22], carrots [22], mangoes [23,24], tomatoes [20,25,26], guavas [27], açaí [28] and even yogurt [29].

Therefore, the purpose of this work was to investigate the potential use of refractance window (RW) drying for processing the microalga *Spirulina platensis*. The effects of operating variables on the moisture removal were evaluated and the quality of the final product, expressed by the content of bioactive compounds (i.e., phenolics, flavonoids, and phycocyanin), were also quantified.

## 2. Results

### 2.1. Visual Aspect

Figure 1 shows the aspect of the fresh and RW-dried Spirulina. Browning and carbonization regions, which are common in conventional methods, were not observed, even in the experiments performed at higher temperatures. This interesting result indicates that this alternative methodology has the potential to process this microalga with minor visual physical damage, similar to what was reported by Nindo and Tang [18] and Abonyi et al. [22] for other biological materials.

### 2.2. Preliminary Tests and Dehydration Kinetics

Table 1 shows the dehydration times, the final moisture content (wet basis) and the water activity (a_w_) obtained in the preliminary tests. As can be seen, the moisture content and a_w_ value of the fresh microalga was 82.7% (wet basis) and 0.967, respectively. The results obtained in these preliminary tests show that the use of RW under those conditions led to a final product with a moisture lower than 15% and water activity lower than 0.600, which represents adequate levels for processing, storage and transportation [30].

Figure 2 displays the kinetic curves obtained for the experiments. The kinetic equation that best represented the experimental data was the Midilli et al. [31]. Table 2 shows the estimated parameters and the correlation coefficient (R^2^) of this kinetic equation. As can be seen, when the temperature increased from 60 to 80 °C, the kinetic constant (k) increased by 168%. This occurred because this parameter (k) is related to the diffusivity of water through the material during drying, which in turn is strongly influenced by temperature [32].

### 2.3. Experimental Design Results

#### 2.3.1. Moisture and Water Activity (a_w_)

A new set of experiments was carried out to evaluate the effects of temperature (T) and dehydration time (t) on the content of bioactive compounds present in the microalga after RW drying. The operating conditions of these tests were chosen using a rotational central composite design (CCD). Table 3 shows the experimental design and the results obtained for final moisture content and water activity (a_w_). As observed, some conditions (i.e., low temperatures and processing times) led to a product with high moisture content and a_w_ levels (e.g., Experiments 1 and 5). The lowest final moisture content and a_w_ were obtained in Experiment 6, which was performed at 84.1 °C for 4.5 h. However, in 70% of the experiments, the values of final moisture content and a_w_ were considered adequate for storage and transportation [30].

#### 2.3.2. Bioactive Compounds

The total phenolic (TPC), total flavonoid (TFC) and phycocyanin (PC) contents after each experiment (see conditions in Table 3) are displayed in Figure 3, Figure 4 and Figure 5, respectively. Regression equations (Equations (3), (5) and (6)) were fitted to the experimental data to quantify the effects of the studied independent variables temperature (T) and dehydration time (t) on the bioactive compounds (TPC, TFC and PC, respectively). From these equations, it was possible to obtain response surfaces (Figure 6) that illustrate these effects.

Analyses of the temperature and drying time effects on the bioactive compounds content have great importance once they can provide important information about the effects of these operating conditions on the quality of the dried product. For example, specific bioactive compounds are highly sensitive to high temperatures and can degrade if the drying process occurs in these conditions. On the other hand, low temperatures can lead to high drying times, which can also cause degradation due to prolonged exposure and add costs to the drying process due to the higher energy consumption [8,15,22,32]. Therefore, an “equilibrium point” between these two variables can help to define the best conditions to operate the RW drying of the *Spirulina platensis* without having an effect on the compounds present in the microalga.

The pharmacological properties of Spirulina, that is, its anticarcinogenic, antiviral, antimicrobial, anti-inflammatory and antitumoral activities, have been directly related to the presence of phenolic compounds [9,12]. Figure 3 shows the total phenolic content (TPC) after each test. As it can be noted, under some operating conditions the dehydrated microalga reached a TPC value close to that observed in the fresh material (462.1 mg gallic acid/100 g of sample in dry matter). For example, after Experiment 8, the TPC value of the microalga was 436.6 mg gallic acid per 100 g of sample (in dry matter). This is a significant result, since some studies in the literature have reported a considerable degradation of phenolic compounds after conventional drying methods [12,33,34]. Therefore, the use of RW drying for Spirulina has the potential to offer a better retention of this compound in the final product.

Equation (3) shows the fitted equation (R^2^ = 0.85) for TPC as a function of temperature (T) and time (t), considering their linear, quadratic and interaction effects. In this equation, as well as in Equations (5) and (6) (for TFC and PC, respectively), the variables are presented in coded form, using Equation (4) below. The variables T and t are in °C and h, respectively.
(1)$$\mathrm{TPC}=337.17+53.27{\;x}_{1}-38.79{\;x}_{1}^{2}+41.95{\;x}_{2}-52.49{\;x}_{1}x_{2}$$
where: (2)$$x_{1}=\frac{T-70.0}{10.0}{\;\mathrm{and}\;x}_{2}=\frac{t-4.5}{1.5}$$

In Equation (3), the positive values of the parameters related to the main effect of the independent variables indicate that at high levels of these variables, the TPC was also high, which can be associated to the presence of melanoidins, as observed from the Maillard reaction [30]. However, there is also a non-linear effect of T ($x_{1}^{2}$) on TPC, which suggests that the highest TPC values are obtained at intermediate levels of temperature (Experiment 8), as illustrated in Figure 6a.

Although the presence of flavonoids in microalgae has been poorly explored in the literature, this bioactive compound has important biological properties, including antioxidant, anti-inflammatory, estrogenic and antimicrobial activities [12,35,36]. Figure 4 shows the TFC observed in the Spirulina samples after drying. As can be seen, the samples had a lower flavonoid content after drying than that found in the fresh microalga (9.86 mg rutin/100 g of sample in dry matter). However, high levels of TFC were obtained under specific operating conditions, such as those used in Experiment 8, which resulted in a TFC value of 7.61 mg/rutin per 100 g of sample (dry matter).

The fitted equation for TFC is shown in Equation (5) (R^2^ = 0.92), where it is possible to note that the independent variables had a similar effect to that observed for TPC. Figure 6b, which was built using Equation (5), confirms the similar behavior of these responses (TFC and TPC) as a function of the independent variables T and t.
(3)$$\mathrm{TFC}=5.64+0.95{\;x}_{1}-0.61{\;x}_{1}^{2}+0.99{\;x}_{2}$$

Phycocyanin is an abundant protein pigment present in the Spirulina biomass that has been commercially used in food coloring and in the cosmetics industry. However, several studies have demonstrated that this compound also has a significant therapeutic value due to its high antioxidant and anti-inflammatory properties, attracting the attention of the pharmaceutical and functional food industries as well [37,38,39,40]. According to Figure 5, the PC value after drying was significantly lower than that observed in the fresh microalga (14.55 g phycocyanin/100 g of sample in dry matter). This decrease was more pronounced in tests performed at higher temperatures, e.g., Experiment 6 (84.1 °C). The thermosensibility of phycocyanin during dehydration has also been observed in other studies using different techniques [6,41,42]. Nonetheless, some operating conditions tested herein led to significant phycocyanin levels in the dried material.

Equation (6) shows the fitted equation (R^2^ = 0.91) for PC as a function of temperature and time. As expected, temperature (T) was the most significant variable, exerting a negative effect on PC. The highest PC was found at the lowest T and t (see Figure 6c), thus confirming the thermosensibility of this compound.
(4)$$\mathrm{PC}=8.25-2.09\;T-0.76{\;T}^{2}-0.92\;t$$

## 3. Materials & Methods

### 3.1. Raw Material

The microalga *Spirulina platensis* used in this study was provided by Brasil Vital, a company located in Goiás State, in the Central-West Region of Brazil. Prior to use, the material was stored in small portions in sealed polyethylene packages and frozen in a freezer (−18 °C).

### 3.2. Experimental Apparatus

The RW lab-scale batch dryer made for this study is presented in Figure 7. It consisted of an aluminum reservoir (28.0 cm × 20.0 cm × 8.0 cm) filled with circulating hot water provided by a thermostatic bath (Tecnal, TE-184, Piracicaba-SP, Brazil), operating in a closed system. The support chosen for this apparatus was Mylar^®^ polyester film type D (DuPont, Wilmingron, DE, USA).

The Mylar^®^ film (with a thickness of 0.25 mm) was fixed on the top of the reservoir frame to ensure that the whole surface of film bottom was touched by the hot water. The water temperature was controlled by an external thermostatic bath (±0.1 °C). For each experiment, about 40 g of fresh Spirulina was spread over the Mylar^®^ film with the aid of a plastic support so as to allow a uniform material thickness of 0.50 cm.

### 3.3. Experimental Design

Preliminary tests were carried out at three different temperatures of hot water (60, 70 and 80 °C) in order to evaluate the moisture removal behavior of this material during RW drying. Once the microalga showed high stickiness and high adherence to the Mylar^®^ film during the drying process, making it difficult the sample collection to the measurement of moisture level during the tests, we choose to spread the Spirulina on aluminum foil, which was placed over the Mylar^®^ film. Desmorieux et al. [33] evaluated the use of aluminum paper as a support during the Spirulina drying, and they showed that this procedure does not significantly affect the results.

After analyzing the preliminary tests, an experimental design was performed to evaluate the effects of operating variables on the content of bioactive compounds. The experiments were organized in a rotational central composite design (CCD) using two independent variables: hot water temperature (T) and dehydration time (t). The coded and real values of the independent variables are shown in Table 4.

### 3.4. Moisture and Water Activity (a_w_) Analysis

The moisture content of the fresh and dehydrated samples (wet basis) was determined by the oven method: 105 ± 3 °C for 24 h, based on the AOAC Official Method [43]. The water activity (a_w_) was measured using aNovasina RS 232/RTD-200 meter (Novasina, Zurich, Switzerland), a porTable water activity device that calculates the a_w_ using a temperature control system integrated to an infrared sensor that provides results with a precision of ±0.001. The water activity of microalgae after dehydration is expected to be lower than 0.600, since in this condition most bacteria, fungi and yeast have their activity and growing inhibited [30,44]. All measurements were performed in triplicate.

### 3.5. Dehydration Kinetics

In general, the dehydration kinetic equations are presented in the form of variation of a dimensionless moisture number (moisture ratio) as a function of time. The moisture ratio (MR) is given by Equation (1):(5)$$\mathrm{MR}=\frac{M-M_{\mathrm{eq}}}{M_{0}-M_{\mathrm{eq}}}$$
where M is the moisture content at any time, M_eq_ is the equilibrium moisture content, and M_0_ is the initial moisture content.

A great number of empirical and semi-empirical equations have been used to describe the dehydration kinetics of biological materials [45,46,47]. Table 5 lists the dehydration kinetic equations used herein, where k, n, A and B are the model parameters. The best equation was selected based on a statistical analysis considering the correlation coefficient, the significance of parameters, and the distribution of residues [48,49].

### 3.6. Analysis of Bioactive Compounds

The content of bioactive compounds in the fresh and dehydrated samples was measured to evaluate the impact of the operating conditions on the quality of the final product. All analyses were carried out in triplicate, and the content of bioactive compounds was expressed as mean value ± standard deviation. All results were evaluated using multifactorial ANOVA. A multiple regression equation was fitted for each response, which was analyzed using response surface methodology.

#### 3.6.1. Total Phenolic Content (TPC)

The total phenolic content was determined by the Folin-Ciocalteau method [54], using gallic acid as a standard and performing spectrophotometric reading at 622 nm (spectrophotometer V1200, VWR, Leuven, Belgium). The results were expressed in milligrams of gallic acid per 100 g of sample (dry matter).

#### 3.6.2. Total Flavonoid Content (TFC)

The total flavonoid content was quantified following the colorimetric method described by Zhishen et al. [55], using rutin as a standard and performing spectrophotometric reading at 450 nm (spectrophotometer V1200, VWR, USA). The results were expressed in milligrams of rutin per 100 g of sample (dry matter).

#### 3.6.3. Phycocyanin Content (PC)

The extraction of phycocyanin was made based on the methodology reported by Costa et al. [8], using water as a solvent extractor and performing spectrophotometric readings at 620 nm and 652 nm (spectrophotometer V1200, VWR, Radnor, PA, USA). The phycocyanin content was obtained using Equation (2), as described by Bennett and Bogorad [56]:(6)$$\mathrm{PC}=\frac{\left[{\mathrm{OD}}_{620}-0.474\left({\mathrm{OD}}_{652}\right)\right]}{5.34}$$
where PC is the phycocyanin content (mg/mL) and OD_620_ and OD_652_ are the optical density of the samples at 620 nm and 652 nm, respectively. The results in all assays were subsequently converted into grams of phycocyanin per 100 g of sample (dry matter).

## 4. Conclusions

The use of refractance window (RW) drying in *Spirulina platensis* biomass proved to be an interesting alternative for the processing and conservation of this microalga. Similar to that observed in other food and biological materials dehydrated by this technique, it was possible obtain a dried product with minimal visual changes and a colour similar to that observed in fresh microalga (Figure 1). The statistical discrimination study of the kinetic behavior revealed that the equation proposed by Midilli et al. [31] was the one that best represented the experimental data (Figure 2).

Although the results indicate that it was possible to satisfactorily remove the moisture of the samples, reaching low levels of moisture and water activity (a_w_), the analysis of the effect of operating variables (temperature and time) on the content of bioactive compounds present in the microalga indicated some interesting results. The effects of this dehydration methodology were similar for total phenolic (TPC) and total flavonoids (TFC) content, with the best results (contents near that found in the fresh microalgae) being obtained at temperatures around 70 °C and processing times of approximately 4.5 h. In contrast, the phycocyanin content was negatively influenced by an increasing temperature due to its thermosensibility properties, reducing its levels in high temperatures (higher than 80 °C) and high exposing drying times (higher than 4.5 h).

Based on these results, we can infer that refractance window drying is an alternative technique that provides an efficient dehydration of the microalga *Spirulina platensis* if performed under adequate conditions. In conclusion, this methodology can be useful for mitigating the negative effects of conservation methods during the processing of this material.

## Figures and Tables

**Figure 1 molecules-28-00720-f001:**
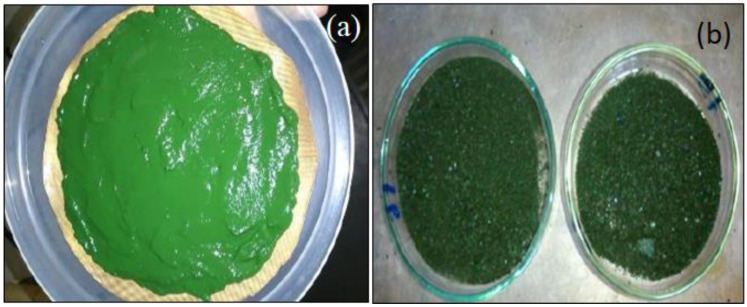
Fresh (**a**) and RW-dried (**b**) Spirulina samples.

**Figure 2 molecules-28-00720-f002:**
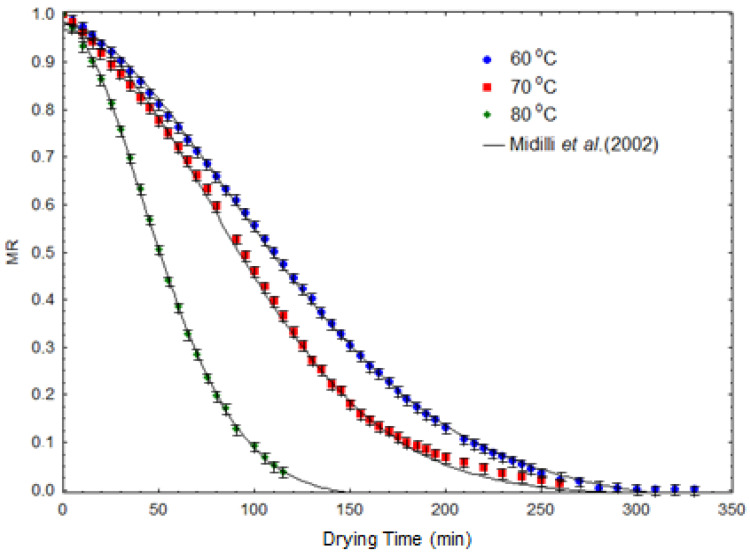
RW dehydration kinetics: experimental results and prediction by the Midilli et al. [31] kinetic model.

**Figure 3 molecules-28-00720-f003:**
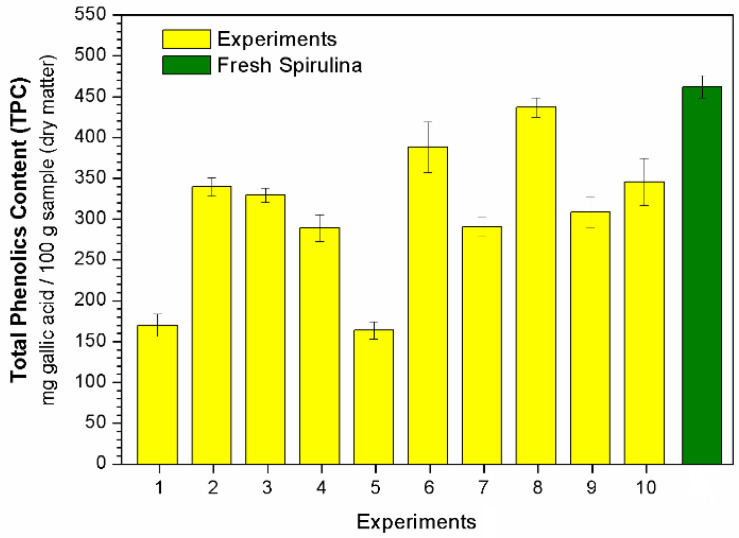
Total phenolic content (TPC).

**Figure 4 molecules-28-00720-f004:**
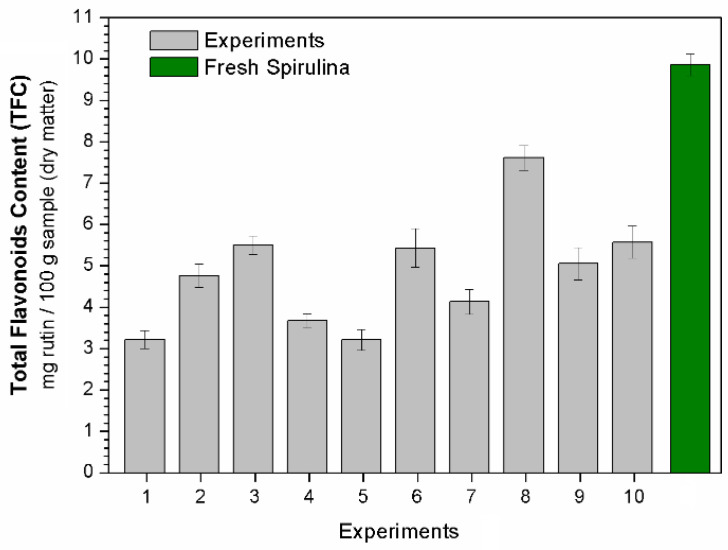
Total flavonoid content (TFC).

**Figure 5 molecules-28-00720-f005:**
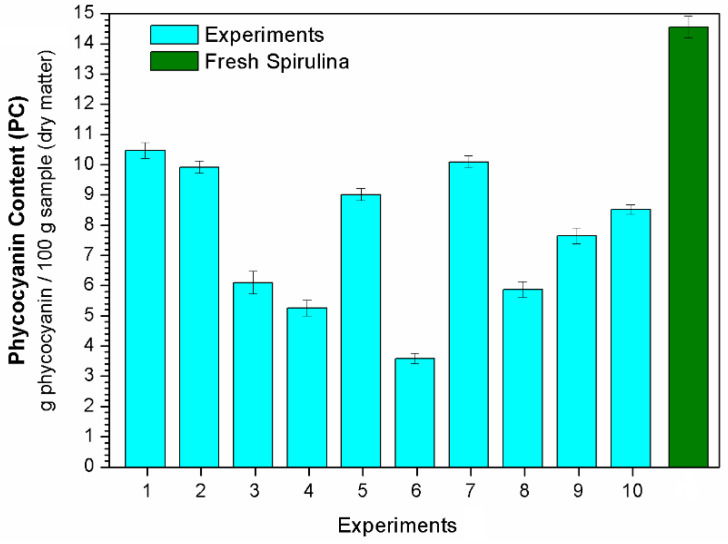
Phycocyanin content (PC).

**Figure 6 molecules-28-00720-f006:**
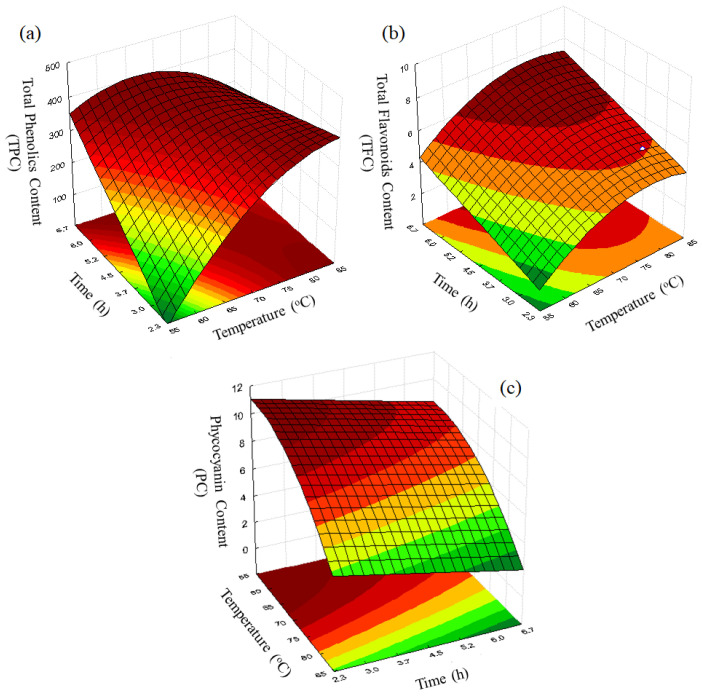
Surface responses for TPC (**a**), TFC (**b**) and PC (**c**).

**Figure 7 molecules-28-00720-f007:**
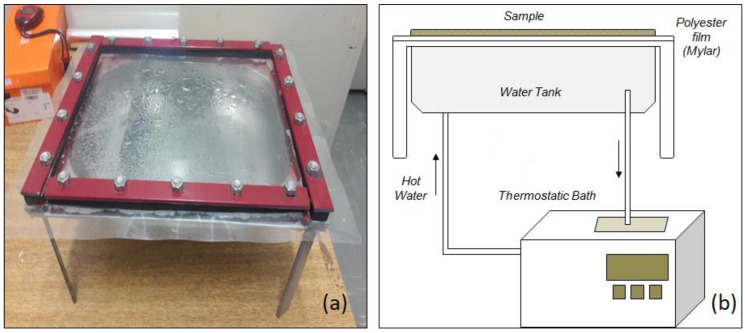
Refractance window (RW) drying experimental apparatus: (**a**) RW dryer; (**b**) schematic Figure.

**Table 1 molecules-28-00720-t001:** Preliminary experimental results.

Exp.	Temperature	Moisture (%)	Water Activity (aw)	Drying Time (min)
1	60 °C	13.04 ± 0.57%	0.588	330
2	70 °C	12.11 ± 0.30%	0.474	260
3	80 °C	10.15 ± 0.07%	0.452	115
Fresh Spirulina		82.70 ± 0.97%	0.967	-

**Table 2 molecules-28-00720-t002:** Parameters of the model proposed by Midilli et al. [31].

Experiment	k	n	A	B	R^2^
60 °C	2.00 × 10^−4^	1.72	0.9782	−8.00 × 10^−5^	0.9996
70 °C	3.42 × 10^−4^	1.77	0.9801	−8.20 × 10^−5^	0.9990
80 °C	5.36 × 10^−4^	1.82	0.9832	−8.60 × 10^−5^	0.9997
				R^2^ medium	0.9994

**Table 3 molecules-28-00720-t003:** Experimental design results.

Exp.	Temp. (T) °C	Time (t) h	Moisture (%)	Water Activity (a_w_)
1	60.0	3.0	30.65 ± 0.16	0.829
2	60.0	6.0	13.60 ± 0.46	0.536
3	80.0	3.0	8.19 ± 0.19	0.367
4	80.0	6.0	8.06 ± 0.19	0.359
5	55.9	4.5	33.72 ± 0.02	0.839
6	84.1	4.5	7.06 ± 0.41	0.339
7	70.0	2.4	21.13 ± 0.51	0.694
8	70.0	6.6	12.17 ± 0.47	0.467
9	70.0	4.5	11.76 ± 0.42	0.418
10	70.0	4.5	10.73 ± 0.08	0.402

**Table 4 molecules-28-00720-t004:** Coded and real values of the experimental design.

Independent Variables	−1.414	−1	0	+1	+1.414
Temperature (°C)	55.9	60.0	70.0	80.0	84.1
Time (h)	2.4	3.0	4.5	6.0	6.6

**Table 5 molecules-28-00720-t005:** Dehydration kinetic models.

Equation in the Literature *	Reference
MR=exp(−kt)	Lewis [50]
$\mathrm{MR}=\exp\left(-{\mathrm{kt}}^{n}\right)$	Page [51]
$\mathrm{MR}=\exp\left[-{\left(\mathrm{kt}\right)}^{n}\right]$	Overhults et al. [52]
MR=A exp(−kt)	Brooker et al. [53]
$\mathrm{MR}=A\exp\left(-{\mathrm{kt}}^{n}\right)+\mathrm{Bt}$	Midilli et al. [31]

* k, n, A and B are model parameters.

## Data Availability

Data supporting reported results are available from the authors.

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
