# Peer review of "Use of Refractance Window Drying as an Alternative Method for Processing the Microalga Spirulina platensis"

_molecules, 2023, doi:10.3390/molecules28020720_

Round 1
Reviewer 1 Report
The work proposes the use of refractance window technology for the processing of spirulina algae with emphasis on the study of drying kinetics and retention of bioactive compounds.
The manuscript is well organized, however, there are some things that can be improved according to the following:
The abstract is very general and quantitative, the work would improve if some values were included.
The abstract makes comparisons with traditional drying methods, however, nowhere in the paper are experiments performed with classical technologies such as hot air drying or freeze drying.
l27 talks about healing properties, what do you mean by that?
It would be interesting to develop more about the properties and composition of spirulina.
l45 What is most commonly used as support are mylar films, as it is written it is incorrect.
Also, it would be good to include more information on the heat and matter transfer mechanisms associated specifically with refractance window technology. Also include information such as temperature ranges, film thicknesses, and mylar thicknesses used.
in 2.1 It is necessary to describe in more detail the filtering operation and the equipment used for this operation. In addition, the wording of 2.1 needs to be improved and the characteristics of the pump used to recirculate the water need to be specified. In addition, Figure 1a does not provide relevant information, review the relevance or improve the figure.
l95 It is necessary to support with some reference the use of the aluminum film.
l106 Information on a standardized method such as AOAC, AOCS, etc. could be included.
Add information on k,a,b constants in table 2.
Many of the analyses do not correctly specify the equipment used in terms of make and model.
What is the basis for the choice of equations/models used? There are more than 20 models commonly used.
The paper could be improved by including calculations of activation energy and diffusivities that would be easy to obtain from the drying kinetics data.
What is the sense of making a visual analysis of the samples, visual analyses are always objective and do not provide useful information in this case. It would be good to consider for example particle size measurement, instrumental color by colorimeter, microstructural analysis just to mention a few.
l175 Clarify what is meant by adequate levels for processing, storage and transport.
l177 If no conventional drying experiments were performed, nothing can be asserted in this regard.
In Figure 3, if measurements were made in triplicate, error bars could be added.
The conclusions are poor, they can be improved and rewritten, for example: L287 is not a contribution as well as when compared to traditional methods that do not correspond to the experiments performed.
Author Response
Below we reproduce the reviewers’ original comments, followed by our responses (R) in italics. Changes in the text in response to the reviewers have been performed using the “Track Changes” function of Microsoft Word, as solicited.
Reviewer #1:
Comments and suggestions for authors:
The work proposes the use of refractance window technology for the processing of spirulina algae with emphasis on the study of drying kinetics and retention of bioactive compounds. The manuscript is well organized, however, there are some things that can be improved according to the following:
- The abstract is very general and quantitative, the work would improve if some values were included.
R: The abstract was modified and quantitative information were included (New Abstract, new page 1)
- The abstract makes comparisons with traditional drying methods, however, nowhere in the paper are experiments performed with classical technologies such as hot air drying or freeze drying.
R.: We agree with the reviewer. The expression “…compared to conventional methods” was removed in the revised version of the abstract (New Abstract, new page 1)
- l27 talks about healing properties, what do you mean by that?
R.: The expression “healing properties” is associated with the properties of bioactive compounds and essential nutrients that can be found in the Spirulina biomass and can be used in the formulation of medicinal and functional foods and supplements. However, we choose to change this expression for “nutraceutical properties” in the revised version of the manuscript (new page 1), that is more adequate to express these properties.
- It would be interesting to develop more about the properties and composition of spirulina.
R.: New information about the properties and composition of Spirulina were now included in the text (new pages 1 and 2)
- l45 What is most commonly used as support are mylar films, as it is written it is incorrect.
R.: This sentence has been corrected (new page 2).
- Also, it would be good to include more information on the heat and matter transfer mechanisms associated specifically with refractance window technology. Also include information such as temperature ranges, film thicknesses, and mylar thicknesses used.
R.: More information about heat and matter transfer RW mechanisms were included in the revised version of the manuscript, as suggested by the reviewer (new page 2).
- in 2.1 It is necessary to describe in more detail the filtering operation and the equipment used for this operation.
R.: The process of filtering was performed by the Brazil Vital company before we acquire the material, and we have not detailed information about that. Once this information is not relevant to explain the biomass characteristics, we choose to remove it, to avoid misunderstanding (new page 3)
- In addition, the wording of 2.1 needs to be improved and the characteristics of the pump used to recirculate the water need to be specified. In addition, Figure 1a does not provide relevant information, review the relevance or improve the figure.
R.: The pump used to recirculate the water is the own pump of the thermostatic bath (Tecnal, TE-184, Brazil). The relevance of Figure 1 is to present in the paper the scheme of the RW system used on this work. We corrected the figure and add a new version of its in the revised manuscript (new page 3).
- l95 It is necessary to support with some reference the use of the aluminum film.
R.: More details about the reference to support the use of the aluminum film were included in the revised version of the manuscript (new page 4)
- l106 Information on a standardized method such as AOAC, AOCS, etc. could be included.
R.: Information about AOAC method was included in the revised manuscript (new page 4)
- Add information on k,a,b constants in table 2.
R.: The information about k, a, b constants was included in Table 2 footnote (new page 5) and in the text (new page 4).
- Many of the analyses do not correctly specify the equipment used in terms of make and model.
R.: The equipment used in the analyses were specified in terms of make and model as suggested by reviewer (new page 5).
- What is the basis for the choice of equations/models used? There are more than 20 models commonly used.
R.: The equations used in this work (Table 2) were chosen based in previous works developed by our research group with biological materials and food residues, that showed the good fit with the experimental data.
- The paper could be improved by including calculations of activation energy and diffusivities that would be easy to obtain from the drying kinetics data.
R.: This is a good suggestion that we will consider in our future works. Thank you.
- What is the sense of making a visual analysis of the samples, visual analyses are always objective and do not provide useful information in this case. It would be good to consider for example particle size measurement, instrumental color by colorimeter, microstructural analysis just to mention a few.
R.: The reason of showing the visual aspect of the fresh and RW-dried Spirulina (Figure 2) was to show that this methodology did not produce browning and/or carbonization regions in the samples that commonly are observed in conventional methods, as observed in the papers of Nindo and Tang [18] and Abonyi et al. [22].
- l175 Clarify what is meant by adequate levels for processing, storage and transport.
R.: The adequate levels for processing, storage and transport are related with the levels of final moisture and water activity. We included this information in the revised version of the manuscript (new pages 4 and 6).
- l177 If no conventional drying experiments were performed, nothing can be asserted in this regard.
R.: We agree with the reviewer observation. This part of the text was modified in the revised manuscript (new page 6).
- In Figure 3, if measurements were made in triplicate, error bars could be added.
R.: Error bars were now included in Figure 3 (new Figure 3, new page 7).
- The conclusions are poor, they can be improved and rewritten, for example: L287 is not a contribution as well as when compared to traditional methods that do not correspond to the experiments performed.
R.: In the revised version of the manuscript, the conclusion section was improved and rewritten, as suggested by reviewer (new pages 11 and 12),
We would like to thank the reviewer for the time spent on reviewing our manuscript helping us improving the article
Related Papers Published in MDPI Journals
Ramírez-Rodrigues, M.M.; Estrada-Beristain, C.; Metri-Ojeda, J.; Pérez-Alva, A.; Baigts-Allende, D.K. Spirulina platensis Protein as Sustainable Ingredient for Nutritional Food Products Development. Sustainability 2021, 13, 6849. doi: 10.3390/su13126849
Bosnea, L.; Terpou, A.; Pappa, E.; Kondyli, E.; Mataragas, M.; Markou, G.; Katsaros, G. Incorporation of Spirulina platensis on Traditional Greek Soft Cheese with Respect to Its Nutritional and Sensory Perspectives. Proceedings 2021, 70, 99. doi: 10.3390/foods_2020-07600
Leiton-Ramírez, Y.M.; Ayala-Aponte, A.; Ochoa-Martínez, C.I. Physicochemical Properties of Guava Snacks as Affected by Drying Technology. Processes 2020, 8, 106. doi: 10.3390/pr8010106
Li, Y.; Aiello, G.; Bollati, C.; Bartolomei, M.; Arnoldi, A.; Lammi, C. Phycobiliproteins from Arthrospira Platensis (Spirulina): A New Source of Peptides with Dipeptidyl Peptidase-IV Inhibitory Activity. Nutrients 2020, 12, 794. doi: 10.3390/nu12030794
Robledo-Padilla, F.; Aquines, O.; Silva-Núñez, A.; Alemán-Nava, G.S.; Castillo-Zacarías, C.; Ramirez-Mendoza, R.A.; Zavala-Yoe, R.; Iqbal, H.M.N.; Parra-Saldívar, R. Evaluation and Predictive Modeling of Removal Condition for Bioadsorption of Indigo Blue Dye by Spirulina platensis. Microorganisms 2020, 8, 82. doi: 10.3390/microorganisms8010082
R: These articles and new comments were included in the revised manuscript, as requested (new references [4, 5, 16, 54, 10]).
Reviewer 2 Report
The manuscript “Use of refractance window drying as an alternative method for processing the microalga Spirulina platensis” has an interesting subject, although the experimental designed is not the appropriate for the process. I consider major revision for the manuscript.
My major concern is in the experimental design. If the authors defined initially the drying kinetics (section 3.2. Preliminary tests and dehydration kinetics), why realize the drying process at 80°C for 6 hours, or 55.9 °C for 4.5 hours (Table 5. Exp 2, 5 etc.) as conditions? The information obtained is not important in terms of the process described before. A kinetics of bioactive compound in function of different temperatures could provide more information about the effect of refractance window drying.
Line 94-95. ” According to Desmorieux et al [26] the presence of an aluminum foil does not impair thermal exchange due its high conductivity” however, in my knowledge, the principal thermal energy transfer in RWD is by radiation. The aluminum foil could avoid the correct thermal energy transfer. Did the authors evaluate the drying process without aluminum foil?
Section 2.1. Raw material. The freezing process could dehydrate the sample. What is the humidity initial of microalga before the freezing?
Section 2.4. Moisture and water activity (aw) analysis. The method used could overestimate the moisture, due thermal degradation process for high temperature.
Figure 3. Improve the caption figure, it should have a short explanation.
The results obtained in the Table 5 present experiments where the moisture equilibrium was not researched in all experiments, which is the intention of do not obtain a dry microalga?
Conclusion section. Lines 286-288. Do the authors test another drying method process to be sure about the time reduction with RWD?
Author Response
Reviewer #2:
Comments and suggestions for authors:
The manuscript “Use of refractance window drying as an alternative method for processing the microalga Spirulina platensis” has an interesting subject, although the experimental designed is not the appropriate for the process. I consider major revision for the manuscript.
- My major concern is in the experimental design. If the authors defined initially the drying kinetics (section 3.2. Preliminary tests and dehydration kinetics), why realize the drying process at 80°C for 6 hours, or 55.9 °C for 4.5 hours (Table 5. Exp 2, 5 etc.) as conditions? The information obtained is not important in terms of the process described before. A kinetics of bioactive compound in function of different temperatures could provide more information about the effect of refractance window drying.
R.: This experimental design was defined with the objective to analyze the effects of the different conditions of temperatures and processing times on the bioactive compounds of the Spirulina after the process of refractance window drying. Thus, we expected obtain conditions as the mentioned by reviewer (80oC for 6 hours or 55.9oC for 4.5 hours) where an overexposure or others with an incomplete drying process can be occurred. These conditions help to understanding the effect of these variables in the degradation or maintenance of the bioactive compound levels during the drying process and thus, they produce an important analysis of the better conditions to process the microalga using this alternative method.
- Line 94-95. ” According to Desmorieux et al [26] the presence of an aluminum foil does not impair thermal exchange due its high conductivity” however, in my knowledge, the principal thermal energy transfer in RWD is by radiation. The aluminum foil could avoid the correct thermal energy transfer. Did the authors evaluate the drying process without aluminum foil?
R.: We did not evaluate the drying process without aluminum foil because the microalga showed high stickiness and high adherence to the Mylar® film after the drying process, which made the sample collection to drying kinetics and to bioactive compounds analyzes very difficult. Desmorieux et al. [30] also used aluminum foil as a material support to spread the spirulina to avoid these effects. We have added new comments in the revised version of the manuscript (new page 4).
- Section 2.1. Raw material. The freezing process could dehydrate the sample. What is the humidity initial of microalga before the freezing?
R.: The initial moisture of the microalga is the same after and before freezing (82.70%) because the Spirulina samples were storage in small portions in a sealed polyethylene package before the freezing process. We included this information in the revised version of the manuscript (new page 3).
- Section 2.4. Moisture and water activity (aw) analysis. The method used could overestimate the moisture, due thermal degradation process for high temperature.
R.: New information on the moisture measure (AOAC standard method) and water activity (aw) analysis was inserted in the revised manuscript (new page 4).
- Figure 3. Improve the caption figure, it should have a short explanation.
R.: The caption of the Figure 3 was improved as suggested by the reviewer (new Figure 3, new page 7).
- The results obtained in the Table 5 present experiments where the moisture equilibrium was not researched in all experiments, which is the intention of do not obtain a dry microalga?
R.: The experimental design used in this work had the intention to analyze the effects of the different conditions of temperatures and processing times in the bioactive compounds of the Spirulina after the process of refractance window drying. Thus, we expected obtain conditions of overexposure or incomplete drying process once these conditions help to understanding the effect of these variables in the degradation or maintenance of the bioactive compound levels during the drying process and thus, they produce an important analysis of the better conditions to process the microalga using this alternative methodology, which considers both aspects, drying performance and the bioactive compounds content.
- Conclusion section. Lines 286-288. Do the authors test another drying method process to be sure about the time reduction with RWD?
R.: This comparison was performed with results observed in the literature and related papers of Spirulina drying. However, the conclusion section was rewritten and this part of has been removed (new pages 11 and 12).
The authors are grateful for the reviewer's valuable corrections that improved the manuscript.
Related Papers Published in MDPI Journals
Ramírez-Rodrigues, M.M.; Estrada-Beristain, C.; Metri-Ojeda, J.; Pérez-Alva, A.; Baigts-Allende, D.K. Spirulina platensis Protein as Sustainable Ingredient for Nutritional Food Products Development. Sustainability 2021, 13, 6849. doi: 10.3390/su13126849
Bosnea, L.; Terpou, A.; Pappa, E.; Kondyli, E.; Mataragas, M.; Markou, G.; Katsaros, G. Incorporation of Spirulina platensis on Traditional Greek Soft Cheese with Respect to Its Nutritional and Sensory Perspectives. Proceedings 2021, 70, 99. doi: 10.3390/foods_2020-07600
Leiton-Ramírez, Y.M.; Ayala-Aponte, A.; Ochoa-Martínez, C.I. Physicochemical Properties of Guava Snacks as Affected by Drying Technology. Processes 2020, 8, 106. doi: 10.3390/pr8010106
Li, Y.; Aiello, G.; Bollati, C.; Bartolomei, M.; Arnoldi, A.; Lammi, C. Phycobiliproteins from Arthrospira Platensis (Spirulina): A New Source of Peptides with Dipeptidyl Peptidase-IV Inhibitory Activity. Nutrients 2020, 12, 794. doi: 10.3390/nu12030794
Robledo-Padilla, F.; Aquines, O.; Silva-Núñez, A.; Alemán-Nava, G.S.; Castillo-Zacarías, C.; Ramirez-Mendoza, R.A.; Zavala-Yoe, R.; Iqbal, H.M.N.; Parra-Saldívar, R. Evaluation and Predictive Modeling of Removal Condition for Bioadsorption of Indigo Blue Dye by Spirulina platensis. Microorganisms 2020, 8, 82. doi: 10.3390/microorganisms8010082
R: These articles and new comments were included in the revised manuscript, as requested (new references [4, 5, 16, 54, 10]).
Round 2
Reviewer 1 Report
The authors of the paper have provided satisfactory responses to the observations and comments. The work has been substantially improved and it is recommended that the work be accepted in its present form.
Author Response
Below we reproduce the reviewer’ original comments, followed by our responses (R) in italics.
Reviewer #1:
Comments and suggestions for authors:
The authors of the paper have provided satisfactory responses to the observations and comments. The work has been substantially improved and it is recommended that the work be accepted in its present form.
R: We thank the reviewer for this positive feedback.

Reviewer 2 Report
I consider the manuscript is suitable for being published. The following suggestions are made for improving the manuscript:
Section 2.2. Experimental apparatus.
For the drying process the thickness of the material to dry is very important. Lines 122 and 123, please, add the thickness of spirulina on the mylar.
Section 2.5. Dehydration kinetics
With the drying kinetics data, another thermodynamic properties could be added, for example, diffusivities and activation energy
The section 3.3.2. Bioactive compounds could be improved with more details of variables processing effect on bioactive compounds. In order to understand the effect of these variables in the degradation or maintenance of the bioactive compound levels during the drying process, an explanation about effect of the temperature or time of drying on bioactive compound must be added.
Author Response
Below we reproduce the reviewers’ original comments, followed by our responses (R) in italics. Changes in the text in response to the reviewers have been performed using the “Track Changes” function of Microsoft Word, as solicited.
Reviewer #2:
Comments and suggestions for authors:
I consider the manuscript is suitable for being published. The following suggestions are made for improving the manuscript:
- Section 2.2. Experimental apparatus: For the drying process the thickness of the material to dry is very important. Lines 122 and 123, please, add the thickness of spirulina on the mylar.
R.: The thickness of spirulina on the mylar was previously related after Figure 1 (New page 3).
- Section 2.5. Dehydration kinetics: With the drying kinetics data, another thermodynamic properties could be added, for example, diffusivities and activation energy.
R.: This work had the intention to analyze the effects of the different conditions of temperatures and processing times in the bioactive compounds of the Spirulina after the process of refractance window drying. However, we will consider this suggestion of the Revciewer in our future works. Thank you.
- The section 3.3.2. Bioactive compounds could be improved with more details of variables processing effect on bioactive compounds. In order to understand the effect of these variables in the degradation or maintenance of the bioactive compound levels during the drying process, an explanation about effect of the temperature or time of drying on bioactive compound must be added.
R.: More details of variables processing effects on bioactive compounds were added in the revised version of the manuscript (New page 8).
The authors are grateful for the reviewer's valuable corrections that improved the manuscript.
